# Comparative Study of Sound Transmission Losses of Sandwich Composite Double Panel Walls

**Chukwuemeke William Isaac \***, **Marek Pawelczyk** and **Stanislaw Wrona**

Silesian University of Technology, Department of Measurements and Control Systems, Akademicka 16, 44-100 Gliwice, Poland; marek.pawelczyk@polsl.pl (M.P.); stanislaw.wrona@polsl.pl (S.W.)

\* Correspondence: chukwuemeke.william.Isaac@polsl.pl

**Abstract:** The increasing motivation behind the recently wide industrial applications of sandwich and composite double panel structures stems from their ability to absorb sounds more effectively. Meticulous selection of the geometrical and material constituents of both the core and panels of these structures can produce highly desirable properties. A good understanding of their vibro-acoustic response and emission index such as the sound transmission loss (STL) is, therefore, a requisite to producing optimal design. In this study, an overview of recent advances in STL of sandwich and composites double panels is presented. At first, some salient explanation of the various frequency and controlled regions are given. It then critically examines a number of parameter effects on the STL of sandwich and composite structures. Literatures on the numerical, analytical and experimental solutions of STL are systematically presented. Efficient and more reliable optimization problems that maximize the STL and minimize the objective functions capable of degrading the effectiveness of the structure to absorb sounds are also provided.

**Keywords:** sound transmission loss; sandwich/composite structures; double panel walls; controlled regions; vibro-acoustic response

## 1. Introduction

The reduction of noise through vibro-acoustic panels has become an interesting area of research in the past few decades. Single and double wall structures have been used for sound insulation purpose and in various applications such as in the aircraft, automotive, marine and building industries [1–7]. The double panel structures (DPS) have gained popularity over their single panel counterparts owing to their better sound insulation properties and industrial applications over a wide frequency range. Moreover, their vibro-acoustic characteristics can still be improved to give better sound absorbing performance. For example, the DPS have been sandwiched with various interlayers and absorbing materials to improve their sound insulation performance [8,9]. Sandwich structures are low weight and high strength structures which consists of two parallel panels, sheets or laminates bonded to a core or having a cavity between them. The panels which are typically known as the skins or face-sheets can be made from different materials such as metallic, composite or their hybrids. Moreover, the cavity or core between the two face-sheets can be architecturally supported or/and filled with some sound insulation materials [4,8]. In this way, optimal double panel structural design can be adopted for different applications. A very important vibro-acoustic index used to characterize sandwich and composite DPS is the sound transmission loss (STL). This is the ratio of the sound power incident on the incident panel to the sound power transmitted by the radiating panel. During the investigation of vibro-acoustic emission, sound waves impact the incident panel, transmit through the cavity/core and eventually emit from the radiating panel as depicted in Figure 1.

The STL of sandwich and composite DPS have been studied by a good number of researchers [10,11]. One of their goals is to study the influence of certain parameters on the STL of DPS [12,13]. These vibro-acoustic structures are being improved upon not only to increase their sound insulation efficiency but also to give them excellent mechanical properties with strong capacity to withstand thermal and environmental stresses. There are two types of geometrical shapes of DPS which are commonly investigated. They include the rectangular or flat plate/shell and cylindrical or curved plate/shell as shown in Figure 1. It should be noted that when the thickness of the panel is far less than the lengths of the other two dimensions, the panel is considered a shell or thin-walled structure [14]. These different panel or shell types can be made lightweight for acoustic application. However, in vibro-acoustic applications, weight reduction produces low STL especially in the low frequency region. The reason for this low STL effect follows the so called mass law which states that the noise being transmitted through a material medium is proportional to its thickness, density and frequency [15]. The challenge, therefore, in recent times is to design and optimize vibro-acoustic DPS that reduce sound more effectively at very low frequency with low density materials. Inspired by this challenge, aeroelastic, viscoelastic and poroelastic materials have been introduced in the core [16–18]; and materials with orthotropic or anisotropic properties have been used as face-sheets [19,20]. Few authors have attempted to improve the transmission loss of DPS at low frequency by attaching mass on the panel [21], in between the two panels [22] and using metamaterials [23,24] on the face-sheets of DPS. A recent research conducted by Sui et al. [25] was based on the use of acoustic metamaterials to obtain high STL of DPS at low frequency.

During noise reduction of sandwich and composite DPS, the effects of geometrical, structural, material and orientated parameters on the STL are predominantly investigated in the low, medium and high frequency zones. This review work, therefore, gives an overview of the various parametric effects on the STL of sandwich and composite DPS. The ultimate goal of this study is to give a synopsis of recent advances in the STL of double wall structures and their up-to-date analytical models, numerical simulation and experimental measurement. In addition, the review seeks to draw the attention of researchers who currently are investigating new methods of improving the STL of lightweight structures in the low frequency regions. A careful understanding of this review will unlock the opportunity of designing novel vibro-acoustic materials capable of reducing noise at any frequency range. The paper is therefore structured thus: first, in Section 2, the STL curves illustrating the frequency types at different controlled regions are presented. In Section 3, different structural materials of DPS used for vibro-acoustic application are discussed. Double panels with enclosed air cavity, sandwich core support and sandwich composite with flat and curved configurations are reviewed. In Section 4, the models and methodologies adopted for obtaining STL of DPS are given. Parametric effects and optimization problems on the STL of sandwich and composite DPS are discussed in Section 5. In addition, a summary of the application and future development of vibro-acoustic materials used for the reduction of noise is presented.

## 2. The Sound Transmission Loss Curve

To analyze fully the noise transmitted during vibro-acoustic excitation of DPS, the STL curve has to be clearly understood. The curve shows the various frequency regions and also gives the actual characteristics of the material to absorb sound. In the STL characteristics curve, several dip frequencies can be observed which may include the resonance frequency ($f_R$), mass–air–mass frequency ($f_{MAM}$), ring frequency ($f_r$), critical frequency ($f_{cr}$) and coincidence frequency ($f_{co}$) as expressed by their formulas in Table 1. For unbounded panels with enclosed air cavity, the first dip of the STL curve corresponds to the mass-air-mass frequency ($f_{MAM}$). Resonance frequencies are commonly observed with flat or rectangular DPS while ring and critical frequencies are characteristics of cylindrical or curved DPS [26]. Coincidence frequencies are common to all considered DPS cases. For flat plate or shell, the resonance frequency ($f_R$) is the first frequency which occurs when the frequency of the vibrating structure reaches its natural frequency. As stated, this frequency is missing in cylindrical shells, rather a ring frequency ($f_r$) which is the first transition frequency is observed as shown in

Figure 1b. At this frequency, the longitudinal wavelength equals the circumference of the structural element [27]. The next observed frequency for the cylindrical or curved shell is the critical frequency ($f_{cr}$). This is the dip frequency where the circumferential wave number and mode order divided by the radius of shell are equal. The expressions of the critical dip frequency of DPS have been given by Blaise et al. [26]. The dip frequency that is common to flat, cylindrical or curved DPS is the coincidence frequency ($f_{co}$). At this frequency, the wavelength of the trace velocity of the acoustic wave becomes equal to the bending wave velocity of the DPS. Moreover, the coincident frequency approaches the critical frequency as the incident angle ($\theta$) becomes 90 degrees. Table 1 gives the formulas used for calculating these frequencies at different regions. From Table 1, the symbols $E$, $R_s$, $\rho$ and $\nu$ represent Young's modulus, radius of the cylindrical shell, density and Poisson ratio, respectively. While $h_s$ and $c$ represent the thickness of the cylindrical shells and velocity of flow in the medium, respectively; the symbols $k$, $h$, $A$, $m_k$, $m_{s1}$, $m_{s2}$ and $H$ are stiffness, thickness, area of the piezoelectric patch of resonator, mass of the equivalent spring, surface mass densities of the two panels and distance between the two panels, respectively. Also, these frequencies are key indicators typically used by researchers to analyze the vibro-acoustic response of double panels or shell structures [28–30].

**Table 1.** Formulas and definitions of frequency terms commonly used at different regions of sound transmission loss (STL) curves.

| Frequency | Symbol | Formula | Definition | Reference |
|---|---|---|---|---|
| Resonance | $f_R$ | $\frac{1}{2\pi}\sqrt{\frac{k}{\rho h A + \frac{1}{3}m_k}}$ | - Frequency at which the vibrating double panel structure (DPS) reaches its natural frequency. Typical to flat panels. | [12] |
| Mass–air–mass resonance | $f_{MAM}$ | $\frac{1}{2\pi}\sqrt{\frac{\rho c^2}{H}\left(\frac{m_{s1}+m_{s2}}{m_{s1}m_{s2}}\right)}$ | - Frequency at which the double panels vibrates on the stiffness of the separating air layer. | [31,32] |
| Ring | $f_r$ | $\frac{1}{2\pi R}\sqrt{\frac{E}{\rho(1-\nu^2)}}$ | Transition frequency where a reduction in STL first occurs especially for curved panels. | [33,34] |
| Critical | $f_{cr}$ | $\frac{c^2}{2\pi h_s}\sqrt{\frac{12\rho(1-\nu^2)}{E}}$ | Dip frequency where the circumferential wave number and mode per radius of shell become equal. | [34,35] |
| Coincidence | $f_{co}$ | $\frac{c^2}{2\pi h_s \sin^2\theta}\sqrt{\frac{12\rho(1-\nu^2)}{E}}$ | - Dip frequency at which the structural and acoustic wave number coincides. | [34,36] |

Several studies of transmission loss for sandwich and composite DPS have revealed various important regions [33]. However, the transmission loss characteristics curve can be generally divided into three regions—the stiffness controlled region, the mass controlled region and the coincidence controlled region [34,37–41] as depicted in Figure 1. From the figures, the stiffness controlled region emanating from the origin to the first resonance ($f_R$) for flat plate or the transition resonance ($f_r$) for cylindrical or curved shell, is seen as a falling STL curve in the low frequency region. The region of the curve between $f_R$ and $f_{co}$ for flat or rectangular structure is the mass controlled region. This curve is typically seen as a rising curve and is controlled by the mass law [42]. In some special cases, for flat or rectangular structure, as the stiffness of the material decreases, a damping controlled region could be observed [43,44]. Moreover, the mass-controlled region is also observed by the cylindrical or curved structures but they are within the $f_r$ and $f_{cr}$ regime as depicted in Figure 1b. The coincidence controlled region is the region of the curve after the coincidence frequency for most flat or rectangular structure. However, this region begins from $f_{cr}$ for cylindrical or curved structures. It could arise when there is a coincidence between the structural and acoustic wavelengths. Apart from these common controlled regions, some authors have described other kind of regions which may exist between the stiffness controlled and the mass controlled regions. For examples, between these regions, Zhang and Du [45]

described the STL of a flat plate to have a few-mode region while Oliazadeh and Farshidianfar [46] described a resonance controlled region of the STL for cylindrical shells.

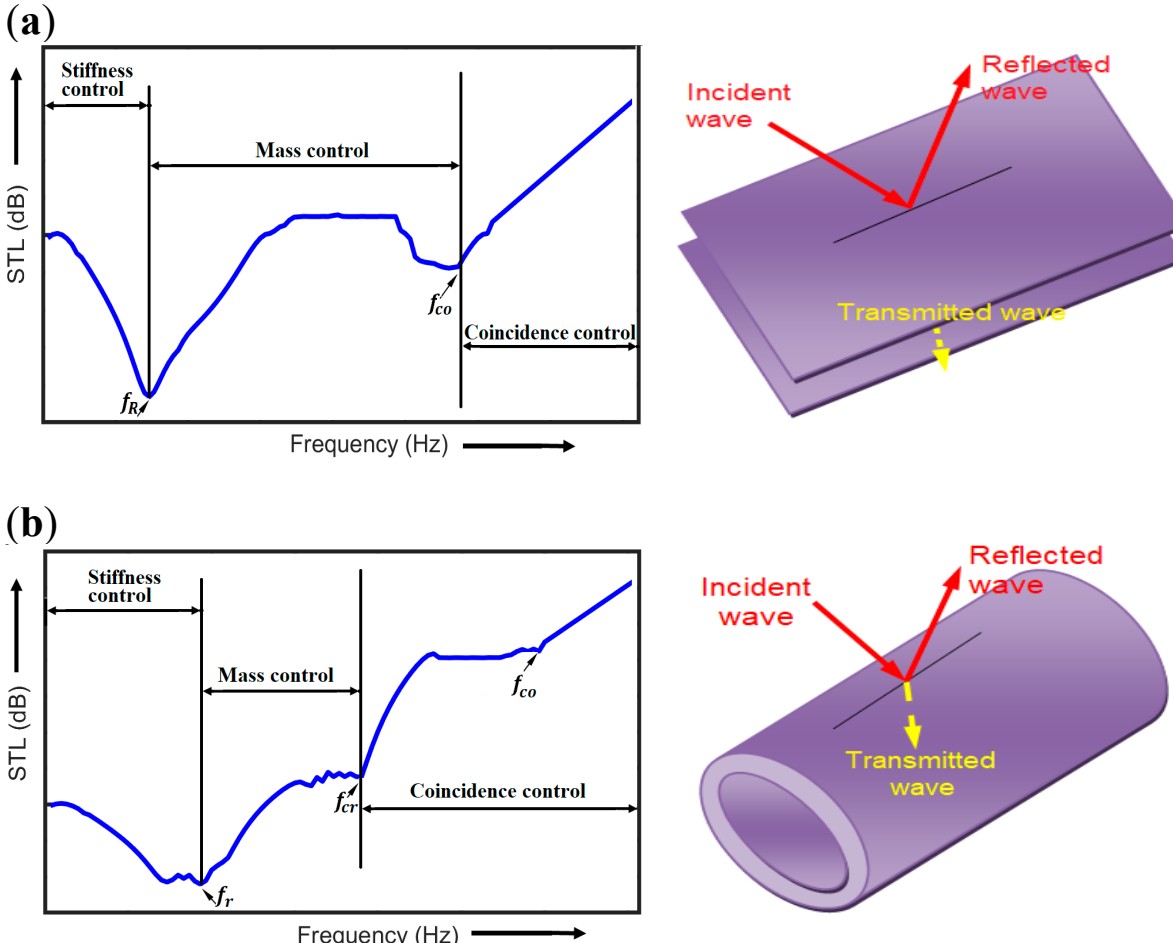

**Figure 1.** Typical representation of sound transmission loss (STL) curve with different frequency characteristics controlled regions for (**a**) flat or rectangular double panel structures (DPS) (**b**) cylindrical or curved DPS.

## 3. Double Panel Air Cavity, Sandwich and Composite Structures

Sandwich structures are generally made of two face panels and low density cores. The materials of the sandwich panels can be made from metals or composites and the cores for example can be made from poroelastic materials such as honeycomb or foams [47,48]. The properties of the sandwich structures that have attracted its usage in vibro-acoustic application are their high strength and flexural stiffness. In this section, different architectural and construction type sandwich panels and cores are discussed.

### 3.1. Double Panel Air Cavity Structures

Double panels with enclosed air cavity may be regarded as the simplest or pseudo form of an equivalent sandwich structures for vibro-acoustic application. Transmission loss results obtained from this equivalent double panel structure can be used as a basis for comparison of the STL results obtained from sandwich core structures. The enclosed air cavity between the two panels allows the transmission of sounds from one panel to the other and has been described to be equivalent to a mechanical spring [49]. It is worthwhile to note that the use of mechanical spring between the panels of the enclosed air cavity or the mass–air–mass (MAM) region of the DPS, reduces their STL especially in the

mid and high frequency region. To validate this statement, the authors of the present study performed a finite element simulation of double panel walls with enclosed air cavity of density $\rho_a = 1.21 \text{ kg/m}^3$ and speed of sound $c_a = 340 \text{ m/s}$. Each panel is simply supported with length $a = 350$ mm, width $b = 220$ mm and thickness $t = 1$ mm. The density, Poisson ratio and Young's modulus of the panels are 2814 kg/m$^3$, 0.33 and 71 GPa, respectively. The effects of spring and damper mechanisms are investigated and the results are compared with double panels without spring and damper effects as given in the finite element results of Figure 2a,b. With spring mechanism connected to the double panels, it is evident that the vibro-acoustic response deteriorates and the response is almost similar to the double panels with enclosed air cavity as shown in Figure 2a. Moreover, a careful inspection of the obtained result shows that at a particular frequency close to the resonance frequency, for the DPS with enclosed air cavity, the two panels move in opposite phase which results in a mass-air-mass resonance ($f_{MAM}$) as shown in Figure 2c. In addition, at resonance dip frequency, the STL of the MAM mechanism is higher than the mass–spring–mass (MSM) double panel mechanism as seen in the STL curve of Figure 3a. In addition, before the resonance dip frequency, it is seen that the STL of the two mechanical structures follow the same trend. The introduction of spring, therefore, increases the stiffness in the enclosed air cavity region. Thus, the result shows that an increment in the stiffness of the cavity or core of DPS undermines the effectiveness of the structure to reduce noise. To improve the STL of DPS with spring connection in the low frequency region, it is necessary to convert the structure to a mass–spring–damper (MSD) system [49]. Figure 3b shows the STL comparison between the MSD system and those obtained from MAM system. It is evident that at some low frequency range, the STL of the MSD system was increased by more than ten percent of the MAM and MSM systems. However, in the mid-high frequency zones (i.e., 900–1000 Hz), there is no improvement of the STL of MSD system and therefore, it can be seen to behave like the air cavity and spring connected DPS.

**(a)**　　　　　　　　**(b)**　　　　　　　　**(c)**

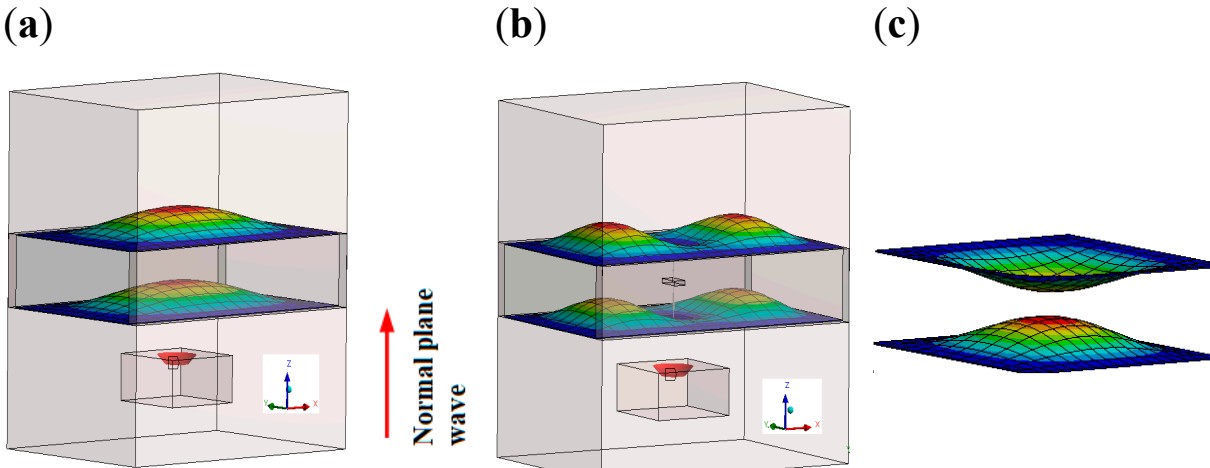

**Figure 2.** Finite element simulation results of double panel enclosed air cavity structure (**a**) first vibration mode of mass–air–mass (MAM) system (**b**) first vibration mode of mass–spring–damper (MSD) system (**c**) vibration mode at MAM resonance.

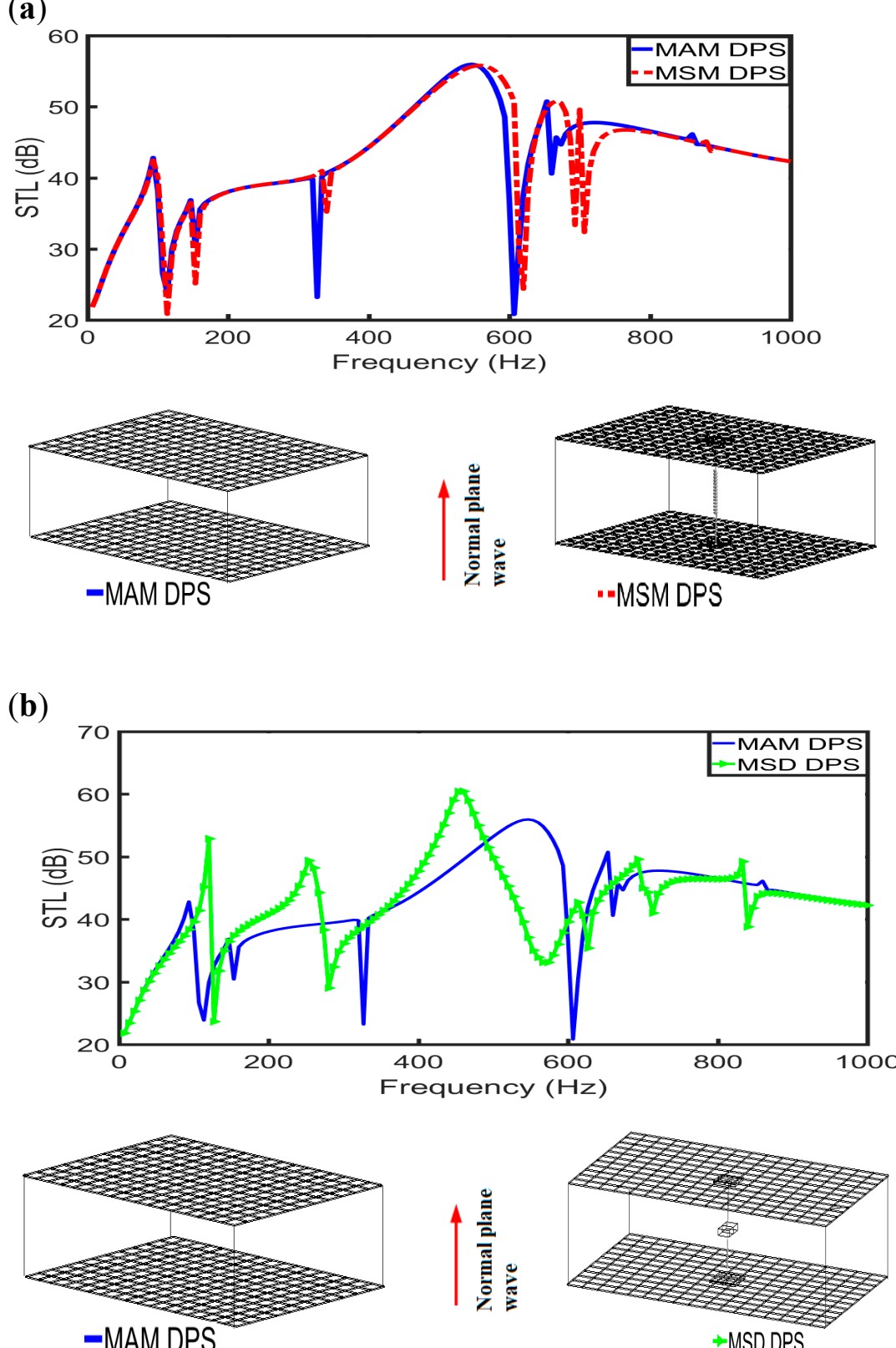

**Figure 3.** Result comparison of double panels and their corresponding finite element mesh configurations for the enclosed air cavity of (**a**) MAM and mass–spring–mass (MSM), (**b**) MAM and MSD systems.

A few research works have been conducted to investigate the STL of DPS with enclosed air cavity [50,51]. Shen et al. [51] compared DPS with air cavity and sandwich structure with corrugated core. In their investigation, it was observed that at some low frequency range, the STL of the DPS with enclosed air cavity followed the same trend with that of the sandwich structure with corrugated core. However, along the medium and high frequency range, the STL of the sandwich corrugated core was lower than the STL of the double panel with air cavity. The reason for this result was adduced to the fact that the corrugated core between the panels provided a strong structural connection that reduced the effectiveness of the sound energy transmitted. To improve the STL of such sandwich corrugated core, an investigation was carried out by Meng et al. [52]. The authors introduced perforations both in the corrugated core and the facesheet panel which significantly increased the STL at the low frequency region. They further showed that panels with non-uniform perforated pore diameter produced higher STL than those whose pore diameters were uniform. Moreover, the corrugated cores were also filled with porous materials and the sandwich plate was upgraded to a functionally graded material. Their solution showed different STL performance of the functionally graded material sandwich structure at different frequency regions. It is therefore necessary to investigate the type of corrugated core and their structural design when it is to be used for specific application to produce improved STL of sandwich structures.

### 3.2. Sandwich Core Support Structures

The core of sandwich structure not only ensures transmission of sounds to the panel but also acts as a mechanical support for the system. Sandwich DPS can be made with different core supports which may include honeycomb [53,54], foam [55,56], corrugated core [57] or pyramidal truss core [58]. Metallic honeycomb core have been widely used in aerospace application [59]. The honeycomb sandwich core provides a bi-directional support for the double panels which in turn gives the entire DPS a high performance to weight advantage. Few studies have revealed the advantage of using honeycomb cores to improve the STL of sandwich structures. Arunkumar et al. [60] carried out a numerical study on the STL of sandwich structures with honeycomb core. In their study, honeycomb cores made from epoxy/graphite fiber reinforced polymer (FRP) were compared with those made from aluminum. Their results indicate that the STL of the former was greater than the STL of the latter below the first resonance frequency. In another study conducted by Li and Yang [47], the STL of double arrowed-head honeycomb cores were investigated and compared with hexagonal honeycomb core by using an increasing and decreasing negative Poisson ratio. Their result showed that the double arrowed head honeycomb DPS with increasing negative Poisson ratio produced better sound insulation performance than the hexagonal honeycomb DPS. Different honeycomb core structures were experimentally investigated by Radestock et al. [61] at low frequency region. The honeycomb core can also be filled with glass fiber or foams such as polyurethane or polypropylene which are commonly used. This combination of honeycomb and foam give the structure better mechanical property. Yang et al. [62] investigated both numerically and experimentally the modal characteristics of cylindrical sandwich structure with foam-filled corrugated core. The curved and flat sandwich corrugated cores are illustrated in Figure 4. Yang et al. [62] studied the STL of honeycomb sandwich DPS filled with glass fiber materials. The fiber materials were filled randomly and in a ball-like structure. The authors investigated their STL with honeycomb sandwich structures that were not filled with fibers. Their results showed a rise in the STL of the glass fiber-filled honeycomb structure as compared to those not filled with fibers.

The sandwich core can be stiffened or structured as a lattice truss core. This 3-D lattice truss core sandwich structure is currently applied in aerospace and marine industries owing to their higher specific stiffness, lower density, lower thermal conductivity and better sound insulation properties. The different forms of these sandwich core supports that have been adopted by contemporary researchers to evaluate STL are the octahedral core, tetrahedral core, pyramidal core, and 3D-kagome core. Wen et al. [63] investigated an octahedral type of core support for the sandwich panel. The authors

cited the advantage of using this particular type of truss core that it has the potential of addressing the acoustic and mechanical properties of the sandwich structure not only in one direction but also in the vertical direction, i.e., two orthogonal directions. The authors also investigated the effects of boundary conditions, core thickness, inclination angle and material parameters on the STL of the octahedral structure and observed significant influence of these parameters on the transmission loss. The tetrahedral, pyramidal and 3D-kagome cores were classified as truss cores by Fu et al. [64]. They performed numerical study on the STL of these truss cores and verified the finite element approximation with a theoretical solution. The effects of various parameters (i.e., truss core radius, height, damping loss factor, external mean flow and material combination types) on the STL of the different truss cores were investigated. The result of their findings showed that at the broad band frequency range, the pyramidal core produced the best sound insulation performance than the other two core types.

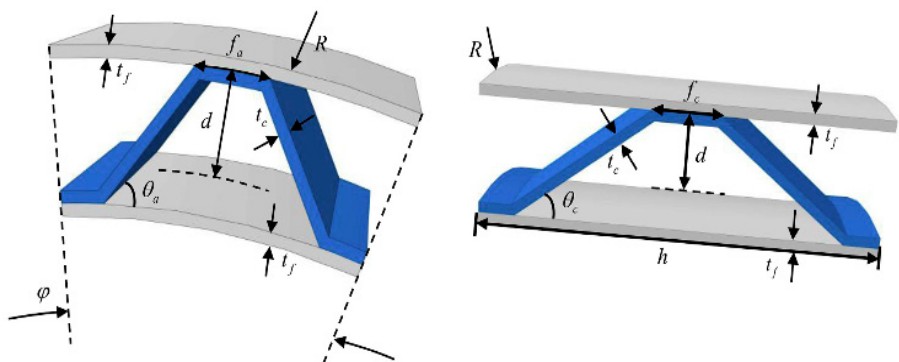

**Figure 4.** Illustration of a unit cell sandwich core support structure for curved and flat plates [48] (reproduced with permission from Yang et al. Polymer Testing; published by Elsevier, 2018).

Wang et al. [65] conducted some experiments to validate their theoretical model on the STL of sandwich panels with pyramidal truss cores. The effects of truss stiffness, incident angles, elevation angles, face-sheet thickness and material properties on the STL of the pyramidal truss core were carefully studied. For example, it was shown that at an incident angle of $0°$, the STL of the pyramidal truss core produced higher results than when incident angles of $30°$, $45°$ and $60°$ were used. In recent time, Wang et al. [31] also developed theoretical model to predict the STL of sandwich structures whose pyramidal truss core are made of two layers and with fiberglass subjected to external mean flow. The influence of various Mach numbers of the external mean flow along the downstream and upstream directions on the STL was compared. In their results, the STL increased with increased Mach number along the downstream direction while it decreased along the upstream directions. In addition, the authors observed minimal shifts in the peaks and dips of the STL curves toward higher frequency during downstream event and toward lower frequency during upstream event. In addition, other parameters such as Young's modulus, fibrous material filling and distribution beams were investigate to evaluate their effects on the STL of the sandwich pyramidal truss cores.

### 3.3. Sandwich Composite Structures

The use of sandwich composite panels with anisotropic properties has been adduced to give higher STL than those with isotropic ones [66]. Sandwich composite structures are particularly suited in the aerospace industry owing to their ability to protect materials and equipment in the aircraft from external noise that could cause their damage. Apart from the good noise reduction behavior of these structures, they also exhibit higher stiffness, higher strength and lower density than their metallic counterpart. The shape of the composite panels investigated by researchers are typically made flat, cylindrical or curved as previously mentioned. However, no sufficient literature has been found to compare the effect of the different geometrical shapes on the STL of sandwich composite structures. For sandwich structures made with metallic skins, Errico et al. [67] proposed a methodology for solving the

STL of periodic flat, cylindrical and curved DPS under aerodynamic and acoustic excitation. Moreover, poroelastic, viscoelastic and aeroelastic materials have been used in the core to further improve their vibro-acoustic properties.

### 3.3.1. Composite Flat Panels

The influence of various parameters on the STL of flat sandwich composite structures as depicted in Figure 5 has been investigated by a good number of researchers [64,68–71]. In recent times, Li et al. [72] performed an analytical solution of the STL of flat sandwich composite structures by expressing the STL as the mode shape superposition. The authors also studied how certain parameters such as the angle of elevation, temperature and azimuth of incident sound influenced the STL. In their findings, it was observed that as the temperature increased, the peaks and dips tend to decrease and flow towards the lower frequency range. In addition, with increased elevation angle, the STL decreased while the effect of the azimuth angle is noticed especially at high frequency region. Further, Wang et al. [73] performed an analysis of sandwich plates whose panels were made of flat composite materials and supported by pyramidal truss core. In their study, they investigated the effect of incident angle, geometrical shape and material properties on the STL of the sandwich composite structures. Their results showed that as the elevation angle of the truss increased, the STL curve became smoother. They also found that the stacking geometry had little effect on the STL in the low frequency region but greater effect on the STL in the high frequency region. In addition, it was observed that altering the mechanical properties of the plates or their structural size will shift and depress their resonance frequency; and when the modulus was also altered, a depression in the extreme values were noticed. An experimental investigation of the vibro-acoustic response of flat, thin and thick sandwich composite panels was performed by Cherif and Atalla [74]. The general laminate model and the equivalent orthotropic panel model were used to predict the STL of the structures. It was found that while the equivalent orthotropic panel model was able to predict the STL perfectly well with the experiment, the general laminate model unexpectedly underestimated the STL due to the possibility of damping effect at coincidence frequency region. The STL due to laminates layup effects of flat composite panels and stiffeners effect between the panels were investigated by Shen et al. [75]. The authors utilized four composite material layers to demonstrate these effects and observed that the STL curve were noticeable varied with the different laminate layup at middle and high frequency regions. Moreover, the STL results were largely dependent of each coincidence frequency of the laminate layup. By altering the laminate layup, the stiffness of the sandwich composite plates were also altered and the position of the coincidence frequencies were shifted. However, in the low frequency region, the effects of laminate layup on the STL were insignificant due to mass inertia of the stiffened structure. Moreover, at this region, the composite panels with stiffened core produced higher STL than those with air cavity owing to mass law effect. However, with increased frequencies, the composite panels were seen to transmit more sound energy than those with air cavity which in turn resulted into lower STL along the mid and high frequency regions. As demonstrated by Qiao et al. [76], it was also shown that the addition of ribs on the radiating face-sheet of flat composite panel has the tendency to lower the STL of the sandwich composite structure. However, in applications where the need arises to include stiffeners on the face-panel of the flat composite structure, it was demonstrated by Shen et al. [75] that by meticulous spacing of the stiffener, the STL can be improved.

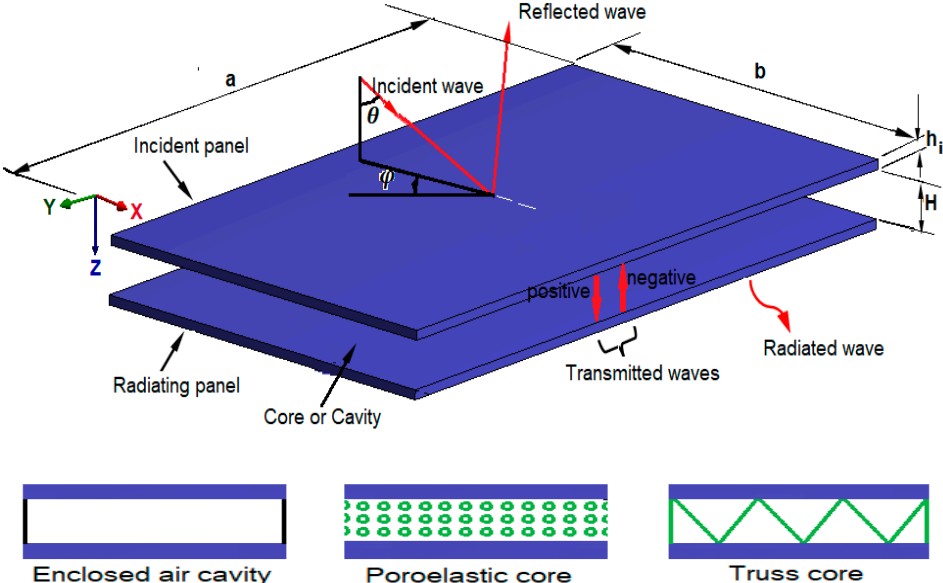

**Figure 5.** Schematic representation of sandwich composite flat panels with different core and cavity types.

### 3.3.2. Composite Cylindrical and Curved Shells

These types of sandwich double wall structures have also attracted attention in the aerospace and marine industry. Their practical application has been seen in the construction of sandwich composite fuselage [59,77]. In the previous subsection, the flat sandwich composite structures were assumed by researchers to have finite structures. However, in the calculation of the STL for the cylindrical composite shells, they are mostly assumed by researchers to be infinitely long. A good number of contemporary authors have investigated the effects of various parameters on the STL of cylindrical sandwich composite shells. Daneshjou et al. [68,78] obtained analytical solutions of STL of composite cylindrical shells and the effect of angle of wrap, shear and rotation of the shells on the STL were in investigated. In their findings, the STL of the composite shells were influenced by the angle of wrap of the layers. However, in the high frequency region, a decreased STL was observed due to the effect of shear wave transmitted but this effect was insignificant in the low frequency region. Talebitooti et al. [69] also analytically obtained the STL of composite cylindrical shell using third order shear deformation theory. The schematic representation of the structure is seen in Figure 6. The authors investigated various parameters such as stacking sequence, Mach number, shell radius, shell thickness, material types and Young's modulus. In their investigation, there was significant effect of the STL when different composite materials were used. The STL evidently increased with shell thickness at low, medium and high frequency zones. Moreover, with increased Mach number and shell radius the STL was seen to decrease especially in the low frequency region. Furthermore, the stacking sequence that involved more zero degree plies were noticed to give better STL. In addition, they also compared the STL results between the composite cylindrical shell and aluminum cylindrical shell. It was seen that the latter gave higher STL than the former in all the controlled regions, i.e., stiffness, mass and coincidence regions. The reason for the higher STL of aluminum cylindrical shell is because it has higher density than their composite cylindrical shell counterpart.

The STL of composite curved shells was also studied by Ghinet et al. [79]. One of the advantages of metallic curved or cylindrical shells over composite curved or cylindrical shells is their higher ring frequency which in turn produces higher STL. Further, another limitation of the composite shells which has affected the lowering of their STL is that their coincidence zone has the proclivity to extend over a wide frequency range than their metallic counterpart. However, insulators, metamaterials and poroelastic materials have been introduced in the cores of composite cylindrical or curved shells to raise their ring frequencies and further improve their STL characteristics. Talebitooti et al. [80] introduced

poroelastic material between doubly curved sandwich composite thin-walled panels. Different design variables such as material type and stacking sequence on their STL were investigated. In their result, it was found that the STL of the poroelastic composite curved shell increased at the medium and high frequency regions. More so, it was observed that at the coincidence dip frequency, the STL of the poroelastic structures were higher than structures without poroelastic cores. Both numerical and experimental investigation were undertaken by Errico et al. [81] to study the effect of foam attachment on the STL of thick sandwich composite curved shell. Two configurations were tested—a curved sandwich panel with attached foam and another without foam attachment. The result of their findings indicated that at frequency above 400 Hz, the STL of the curved panel with foam were higher than those without foam. This foam attached curved panels were also tested using resonators to improve their STL at the ring frequency [82].

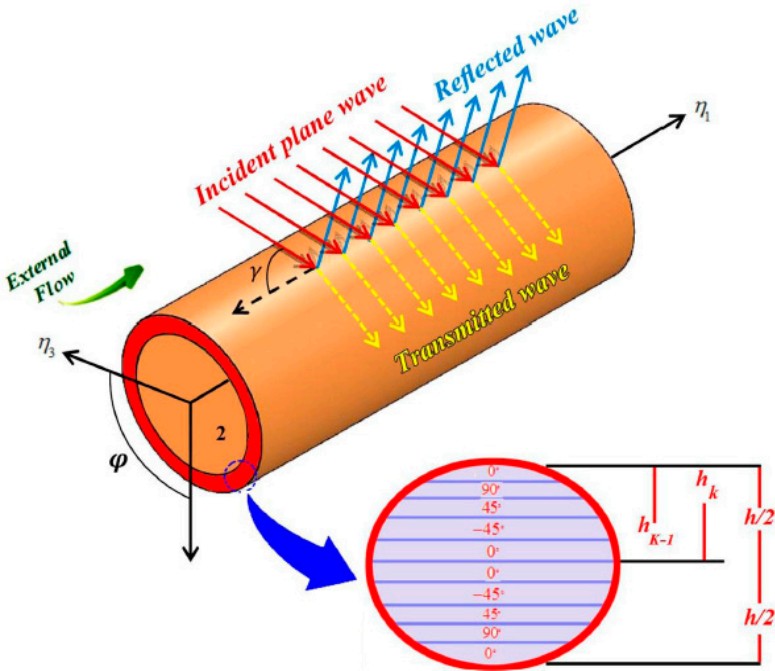

**Figure 6.** Schematic representation of sandwich composite cylindrical shell [69] (reproduced with permission from Talebitooti et al. Composite Structures; published by Elsevier, 2016).

### 3.4. Functionally Graded Materials of Sandwich Structures

Functionally graded materials (FGM) are materials whose composition or microstructures are graded and patterned to give specific functions. While the conventional composite laminates have sharp interface between their constituent materials (i.e., as shown in Figure 7b) which makes them susceptible to high stress concentration, the FGM, however, have smooth variation at their interface as shown in Figure 7a. As a result of this, their properties gradually change in the unit cell, therefore, giving them strong bonding and reduced stress concentration [83]. The vibro-acoustic response and STL of this type of material was studied by Chandra et al. [84]. The rectangular panel was graded with metal (i.e., aluminum) and ceramics (i.e., alumina and zirconia). The effects of incident angles with different power law indices on the STL were investigated. Their results before the resonance frequency indicated higher STL of the ceramic-rich graded material over the metal and metal-rich graded material. Moreover, at the resonance dip frequency, the STL curve of the ceramic-rich FGM shifted to a higher frequency than the metal and metal-rich FGM for all incident angles investigated. A three dimensional analytical model was developed by Daneshjou et al. [33] to estimate the STL of a FGM cylindrical shell as shown in Figure 8. The graded constituents were also metal (i.e., aluminum) and ceramics (i.e., alumina and zirconia) as material of choice previously used by Chandra et al. [84].

Their result indicated enhancement in STL with judicious material distribution through the thickness of the FGM. In addition, the alumina was observed to be mostly efficient in the stiffness controlled zone while the zirconia was mostly efficient in the mass controlled zone due to mass law effect. It should be noted that for thin-walled cylindrical structures, increase in their densities causes the STL to increase especially in the mass controlled zone. However, for thick-walled cylindrical structures, increased densities do not necessarily increase the STL because during transmission loss of the structure, the mass controlled region disappears [85]. Moreover, the authors [33] took into consideration the external flow effect and various Mach numbers from −0.5 to 0.5 were studied. It was observed that the STL increased with Mach number at stiffness and coincidence controlled zones especially when the Mach numbers were negative. To further study the effect of some parameters on the STL of double panel with FGM, an analytical investigation was performed by Danesh and Ghadami [86]. The structure consisted of an enclosed acoustic cavity with double-wall FGM piezoelectric plate. The power law distribution was used to gradually alter the material properties in the direction of the thickness of the plate. In the aforementioned double wall investigations, only the panels or shells were made with FGM. However, the core of the DPS can be made of FGM as demonstrated by Chandra et al. [87]. Other notable authors [88–92] have adopted the FGM to model the vibro-acoustic response of structural panels.

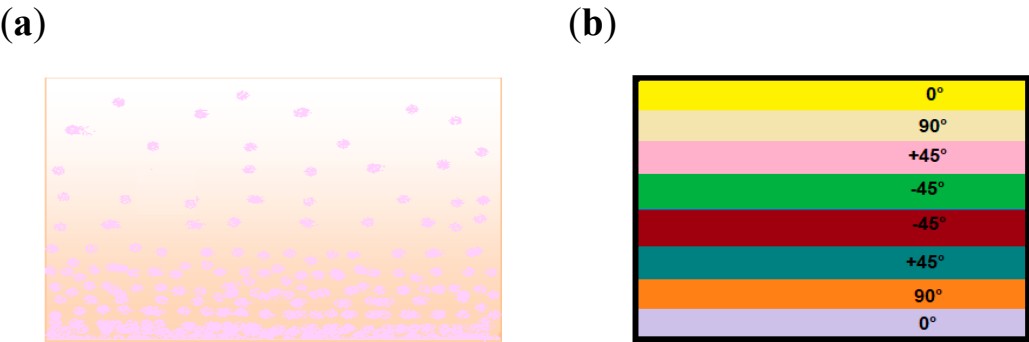

**Figure 7.** Comparison between (**a**) functionally graded material with smooth graded variations and (**b**) composite laminates with sharp interfaces.

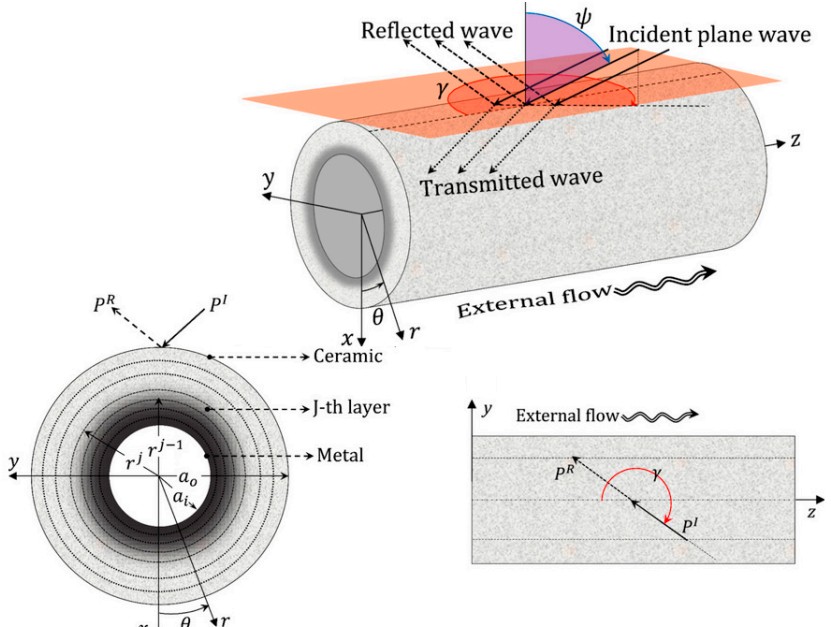

**Figure 8.** Functionally graded multi-layered cylindrical/curved wall [33] (reproduced with permission from Daneshjou et al. International Journal of Mechanical Sciences; published by Elsevier, 2017).

## 4. Methodologies and Models for Obtaining Sound Transmission Loss of Double Panel Walls

*4.1. Models and Methods for Sound Transmission Loss Calculation*

Over the years, quite a number of numerical and mathematical models, theories and methods have been developed by researchers to describe vibro-acoustic response and calculation of the STL of sandwich and composite DPS. For example, the DPS with corrugated cores investigated by Bartolozzi et al. [93] was described by using the Timoshenko beam theory for the sandwich panels while the coupling between the core and the faceplate was described using the classical lamination theory. Some selected theories and models are herewith discussed.

### 4.1.1. Statistical Energy Analysis

This statistical method describes the energy flow correlation for the vibration modes of resonating DPS. It has been suitably adopted when dealing with large structures and vibro-acoustic excitation with high frequency response. The prediction of the STL of sandwich structures using the statistical energy analysis (SEA) method was studied by Zhou and Crocker [94]. In their study, the predicted results of the honeycomb sandwich structures filled with foams were compared with the measured data. The authors stated that the accuracy of the predicted solution of the STL was dependent of the accuracy in the estimation of some parameters such as the modal density, internal and coupling loss factor of the structure. The predicted and experimental solutions showed good agreement near the critical frequency when the values of the radiation loss factor were used rather than the theoretical ones used for single layered panel. In the same vein, Wang et al. [95] predicted the STL sandwich structures using SEA. Their approach took into consideration antisymmetric and symmetric movement of the panels which are calculated independently. The authors' predicted solution showed improvement when compared with the conventional SEA solution and it also matched well with the measured data. Other authors [79,96–98] have adopted this approach to predict the STL of double walls, composite and sandwich structures.

### 4.1.2. Transfer Matrix Method

This method (TMM) can be used to model acoustic waves passing through a double wall structure such as the cylindrical wall as described by Parrinello et al. [99]. The calculation is based on the relations between displacements and stresses at the interface of each layer. The method can be used to model multi-layered noise control treatment for example, in functionally graded material and composite laminates. Daneshjou et al. [33] employed this method to estimate the STL of a functionally graded cylinder under subsonic external flow. Effects of certain parameters such as the Mach number, incident angle, panel thickness and radius of the cylindrical structure on the STL were investigated. Their study showed that the model can predict the STL of multi-layered structures in the low, medium and high frequency regions. Liu and He [100] adopted the TMM to solve the vibro-acoustic problem of composite cylindrical shells with poroelastic core under external mean flow. The Love's theory was used to model the shells while the Biot's theory as will be explained in the subsequent subsection, was used to model the poroelastic core. By employing the transfer matrix formulation with suitable boundary conditions, the solution of the system was obtained. Three different configurations were tested using this method. The configuration whose wave amplitude and the shell displacement were obtained from the derived six shell equations and four boundary conditions, gave the least optimal solutions. However, the STL was enhanced for the configurations with additional boundary conditions due to the presence of air cavity gap in the core. Moreover, the STL of the three configuration lined with poroelastic material was higher than those without poroelastic ones. Two forms of TMM are the impedance matrix model and superposition model. A comparison between these two models on the transmission loss was performed by Hua et al. [101]. It was observed by the authors that the superposition method was better than the impedance matrix method because for the superposition method, matrix assembling was not required for obtaining the transmission loss.

### 4.1.3. Discrete Laminate Model

The general discrete laminate model (DLM) as described by Ghinet et al. [79] with Ghinet and Atalla [102] combines the attributes of discrete sandwich model and the symmetric laminate model. While the former assumes thin laminate skins and a shear bearing core, the latter assumes each laminate layer thick. The use of DLM permits both the skins and core to have thick laminates such that each laminate layer thickness has smeared features. Moreover, it can handle both the symmetrical and asymmetrical behavior of composite and sandwich shells having thin or thick laminate skins.

### 4.1.4. Wave Spectral Finite Element Model

Mejdi et al. [103] develop a wave spectral finite element model (WSFEM) which is based on the DLM. Their model permits the prediction of STL of sandwich structures whose cores are thin or thick as well as stiff or soft. The stretch and compressive components in the layers were described using the properties of the layer for a forced trace wave number and heading direction. In their study, it was observed that for a small flexural wavelength without boundary condition effect, the model can be used to reproduce the response of sandwich panels that are simply supported. More so, the model can be improved to predict the response of sandwich configuration to control noise effects which are particularly excited by turbulence and mechanical load.

### 4.1.5. Wave Finite Element Method

Finite element methods have been used to model the vibroacoustic emission of sandwich and composite DPS [32,104]. An improved method—the wave finite element method (WFEM) described by Mace and Mancony [105], Droz et al. [106] and Chronopoulos et al. [107], is a numerical model reduction method which permits the modelling of each cell element rather than modelling the complete structure. The method entails the combination of both the finite element and the periodicity of the structure with the advantage of low computational cost. In recent times, this model was adopted by Yang et al. [108,109] to predict the STL of multi-layered panels with fluid layers. In their approach, small segments of the solid layers were initially discretized to evaluate their stiffness and mass matrices. Different conditions were applied to determine their spectral dynamic stiffness matrices and the acoustic response of the panels during excitation. The effect of finite size on the STL was investigated and the authors arrived at the conclusion that the WFEM was able to predict the STL very fast with low computational cost. Zergoune et al. [110] also adopted this model recently to model the STL of sandwich structures particularly in the low and medium frequency range. The authors used the DLM and the WSFE model with experimental results to verify and validate, respectively their predicted model. It was evident from their results that while the cell angle of the hexagonal core had no influence on the STL, increased thickness of the panels increased the STL and a lowering of the stiffness-weight ratio in the mid frequency regime.

### 4.1.6. Biot's Theory and Equivalent Fluid Method

Periodic structural links when applied appropriately between the sandwich panels are very effective in evaluating the STL at low and mid frequency regions. Various models for periodic structural links have been examined by Legault and Atalla [111]. Moreover, the introduction of porous media or poroelastic materials in between these structural links of sandwich and composite panels can enhance sound reduction in the mid and high frequency regions. Two basic important theories have been used to model these poroelastic materials; they are the Biot's theory [112–115] and the equivalent fluid method (EFM) [36,116,117]. While the Biot's theory assumes an elastic frame in the porous media such that there exist two dilatational or longitudinal (i.e., compressive) waves and one rotational (i.e., shear) wave; the EFM assumes that the porous media is equivalent to a fluid medium having density and bulk modulus. A comparison between these two approaches for fiberglass and foam cores were

studied by Panneton and Atalla [118]. In their analyses, the two approaches not only represented well the STL but also gave similar predictions.

### 4.1.7. Hybridization Models

A combination of two or more models to predict the STL of sandwich or composite DPS can result into a hybridized model. One possible reason for hybridization is to take care of limitations which could arise from a single model adaptation. For example, Decraene et al. [119] combined the TMM and the hybrid finite element (FE)-SEA models to give a hybridized TMM-SEA model. In their analysis, two disadvantages of the single TMM were listed. The first is that the TMM model requires high computational time to integrate the transmitted wave over all the possible incident angles. Further, the second is that they are very limited in the computation of the variance of STL. While the hybrid FE-SEA model [120,121] was able to handle the shortcomings of TMM by computing effectively the mean and variance at low computational cost, they however, are computationally burdensome at higher frequency range. This is because; a very fine mesh is required at high frequency. The limitations of TMM and FE-SEA therefore, led to the authors' [119] hybridized model. Their model enabled the computation of the mean and variance of diffused field STL and also avoided the detailed finite element modelling of the structure. Other hybrid models which have been investigated are the finite element transfer matrix method (FE-TMM) [122,123] and the patch transfer models [124,125].

### 4.2. Numerical Methods and Tools for Simulating STL

As discussed in the previous sections, a good number of researchers have used analytical tools to predict the STL of sandwich and composite double wall structures. Analytical solutions give the exact solution of the physical problem. However, representation of complex configurations using analytical approach may be intractable. Motivated by this challenge, numerical methods such as finite element methods are better suited to model complex scenarios. The numerical solutions are approximation solutions while the exact solutions can be used to verify these approximations. In addition, the numerical approach not only produces interesting results despite the complexity of the structure, it can also be used to virtually validate analytical and experimental results. A number of numerical tools have been utilized to obtain solutions of sound transmission in different vibro-acoustic media. Typical of these tools are the Comsol Multiphysics [126–129], Ansys Acoustic (ACT) [70,130–132] and Abaqus [133,134]. However, the use of Abaqus software for STL solution has been restricted to lower range of frequencies. This is because; it is computationally burdensome especially in the high frequency range [135]. The following sections discuss a combination of STL solutions undertaken by researchers for verification and validation of their research findings.

### 4.3. Numerical Vs. Analytical Verification

A very few attempts of using mathematical solution to verify numerical approximations have been carried out by researcher. Legault and Atalla [136] coupled insulators in the DPS and evaluated the sound transmission using finite element simulation. The finite element approximation was compared with the theoretical periodic model and it was seen to be well reproduced with the periodic approach. Yuan et al. [71] predicted the STL of fuselage structure using analytical approach, SEA and hybrid FE-SEA methods. The fuselage structure was designed as a double panel cylindrical composite shell with infinite length. Various parameters such as the structural dimension and morphology, material fiber and laminate layup were investigated. In addition, different epoxy laminates such as fiberglass, graphite and aramide were compared. In their comparison, it was shown that the fiberglass/epoxy laminates gave the highest STL especially in the mass-controlled and damping-controlled regions. Moreover, the three methods produced a good representation of the STL at low, medium and high frequency range. Song et al. [137] numerically studied the STL of sandwich plate using the stop band concept. A stepped resonator was attached to the sandwich plate and was compared with sandwich plate without stop band. Their result analyses showed that the STL of the configuration with stop band

was increased around the entire stop band and in some places outside the stop band. The authors also adduced the enhancement of the STL with stop band to mass density increment and changes in the wave number of the plate. The addition of resonators on the composite face-sheet can also be used to generate the stop band. Oyelade [138] theoretically modelled the sound transmission of DPS with magnetic connection. A numerical simulation of the STL was also performed using the COMSOL multiphysics. Comparison was made between the two solutions. In his findings, the STL was affected due to the magnetic stiffness especially for the finite DPS. However, it was also observed that the STL was not affected when an infinite plate was used. In addition, the STL can be improved for DPS with air cavity by connecting the two panels with negative stiffness rather than positive stiffness.

### 4.4. Numerical Vs. Experimental Validation

The various researchers who modelled the STL through numerical methods also conducted some experiments and the results of the measured data were compared with results from the numerical approximation. To account for complexities like perforations on facesheet/panel and core, a numerical study was performed by Meng et al. [52] and their findings were validated by measured data. Further, Kim et al. [139] numerically predicted the STL of complex extruded panels. They applied the wave number domain numerical method to predict the STL of panels with constant cross-section in one direction. The predicted solutions were compared with measured results and a fair correlation of the two results was observed. A simply supported boundary condition and fixed boundary condition were given to the structure. It was observed that the different boundary conditions have little influence on the diffuse field STL. In similar vein, the STL was not much affected with the effect of internal air between the opposite plates due to the strong coupled internal stiffeners applied between the outer plates. The authors, de Melo Filho et al. [140] developed the finite element based unit cell to model the STL of metamaterial double panel at the mass-air-mass resonance frequency region. Their predicted approximations were compared with measured data for validation. Their results showed that STL of the predicted solution and the insertion loss of the experimental results were increased at the targeted mass–air–mass resonance frequency range than the original DPS without metamaterial. The effect of side walls through sonic crystal was investigated by Gulia and Gupta [141]. The authors also compared the experimental insertion loss with the numerical predicted STL. Their results indicated that the best STL was observed when the outer wall of the sonic crystal located at the point where the half of the periodic distance from center position of the end scattered. To further improve the STL, bi-periodicity was introduced.

### 4.5. Analytical Vs. Experimental Validation

A handful of research works on the STL of sandwich and composite structures have been carried out using analytical solutions and these have been validated through experimental measurement [142,143]. Most authors validated their analytical predictions by either performing experiment using the same parametric conditions or by comparing them with the measured data of previous works. Oliazadah and Farshidianfar [46] performed experiments to validate their analytical solution of the STL of cylindrical shells with absorbing materials. The SEA method was used to obtain exact solution while two experimental techniques were used to obtain the measured data. The experimental methods utilized were the transition suite and sound intensity methods. The results of their investigation showed that the analytical solution used for predicting the STL gave improved results especially in the low frequency region due to the application of correction factors at this frequency range. Moreover, a good correlation of the STL at the ring and critical frequencies were observed for both the exact and measured solutions.

## 5. Parameter Effects and Optimization Strategy on the Sound Transmission Loss of Double Panel Walls

### 5.1. Influence of Various Parameters on the Sound Transmission Loss of Double Wall Structures

Parameter effects on the STL of acoustic panels/shells have been used by most authors to determine their noise absorbing efficiencies. For example, Fu et al. [64] obtained interesting results of various parameter effects on the STL of sandwich truss cores. The authors expanded a full analysis of some parameter effects on the STL of three truss cores which included the pyramidal, 3D-kagome and tetrahedral cores. The pyramidal truss core was reported to give the best STL and was used as a standard for investigating the influence of other parameters. This is because, the STL of the pyramidal truss core at the dip and peak frequencies were higher than the 3D-kagome and tetrahedral cores. Structural parameters such as different damping loss factors and different Mach numbers; as well as geometrical parameters such as different core radii and different core heights were studied. First, different material composition types of the flat panels were compared. From the result of their findings, the pyramidal core gave the highest STL over other truss core types at all dip frequencies and its curves shifted to lower frequencies. With an increasing damping ratio, the authors reported no significant increase in STL at all frequencies except at the resonance dip frequencies. At these dip frequencies, the dips flattened and moved upward with increasing damping ratio. On the contrary, the effect of different Mach numbers at all resonance dip frequencies was insignificant; however, there were significant differences in STL at the non-resonance frequency (NRF) regions. At these NRF regions, the STL increased with decreasing Mach numbers. In addition, different material compositions of the sandwich panels resulted in various magnitude of STLs in the resonance or non-resonance frequency regions. With different core radii, the frequency curves shifted to lower frequencies but the STL increased with increasing radius at the NRF regions. Finally, the STL increased at dip frequencies with increasing core and the frequency curves was shifted to higher frequencies.

Apart from the parameter effects on the STL of the acoustic structure performed by Fu et al. [64], a number of parameters have also been tested by different researchers. They can be grouped into geometrical parameters, structural parameters, material parameters and orientated parameters, as represented in Table 2. The table gives a quick overview of the parameter effects on STL studied by researchers in recent times. For example, in the structural parameter group, different sandwich enclosed gas cavity types were investigated by Danesh and Ghadami [86]. The six enclosed gas cavity types were oxygen, air, argon, neon, helium and hydrogen. Their findings evidently depicted a significant effect of STL of the different cavity gases used especially in the low frequency region. While hydrogen and helium gases showed the highest STL at this region, the air and oxygen cavity gas types showed the least STL. The reason for this was attributed to the fact that the speeds of sound in hydrogen and helium gases are higher than the speed of sound in air. In the same vein, Oliazadeh and Farshidianfar [46] investigated the effects of these gases on the STL of cylindrical shell. Their results verified those performed by [86] in that, the hydrogen and helium gases produced the highest STL results than other gases in the low and mid frequency domains. To further explain the grouping of Table 2, some orientated parameters such as the incident angles and stacking sequences were investigated by Talebitooti et al. [69]. With increased incident angles, the STL in the low and high frequency range decreased while there is no change in STL in the mid frequency mass controlled region. Moreover, for the stacking sequence, the authors observed that the STL was enhanced by using plies with more zero degree ply orientation. In the same vein, three orientated parameters such as temperature, elevation angle and azimuthal angle were studied by Li et al. [72]. Their findings showed that at low and mid frequency regions, the STL decreased with increased temperature. In addition, while the STL decreased at low frequency region with increased elevation angles, the increased azimuthal angles showed no significant STL increase at this frequency region. However, at mid-high frequency region, the STL was described as being complex with increased azimuthal angles. From the table, it is evidently seen that many of the material and orientated parameter types have little

influences on the STL. On the other hand, the geometrical and structural parameter types are seen to have great influences on the STL of sandwich and composite double wall structures.

**Table 2.** Summary of selected studies of parameter effects on STL of sandwich and composite double panel structures.

| Group | Parameter | Reference | Influence on STL | | | |
|---|---|---|---|---|---|---|
| | | | Frequency Region | | | Remark |
| | | | Low | Mid | High | |
| Geometrical | Core heights | [51,64,71] | ↑ | ⟺ | ⇑ | - Increased STL. |
| | Core radii | [64,69] | ↑ | ⟺ | ⇑ | - Increased STL. |
| | Cylindrical Shell radii | [68,69,86] | ↓ | ↓ | ⋯ | - Decreased STL with higher radius. |
| | Plate/shell thicknesses | [51,69,144,145] | ↑ | ⟺ | ⇑ | - Increased STL. |
| | Cylindrical shell lengths | [145] | ↓ | ↓ | | - Decreased STL. |
| | Finite dimension of panels/shell | [51,146] | ⋯ | ⋯ | ⇑ | - Lower STL than infinite dimension. |
| | Infinite dimension of panels | [51,146] | ⋯ | ⟺ | ⇑ | - Higher STL than finite dimension. |
| | Different boundary conditions | [13,145] | ↑ | ⟺ | ⋯ | - STL most significant at low frequency regime. |
| | Different core topologies | [70,147] | ↑ | | | - Significant difference in STL. |
| Material | Mach numbers | [11,64,69,100] | ↓ | ⟺ | ↓ | - Poor STL. |
| | Damping loss factors | [64] | ⋯ | ⋯ | ⋯ | - STL decreased; however, it increased at dip frequencies only. |
| | Young's modulus | [31,63,69] | ↑ | ↓ | ⇑ | - Improved STL. |
| | Shear modulus | [29,69] | ⋯ | ⋯ | ⋯ | - No significant difference in STL. |
| | Poisson ratios | [69] | ⋯ | ⋯ | ⋯ | - No significant difference in STL. |
| | Stiffness | [69] | ↑ | ↓ | ⇑ | - Improved STL. |
| Structural | Cavity gas types | [46,86] | ↑ | ⟺ | | - STL largely influenced. |
| | Core types | [31,70] | ↑ | | | - STL largely influenced. |
| | Porous and fibrous material in the core | [118,148] | ↑ | ⟺ | | - Higher STL than cores without porous material. |
| | Different metallic plate/shell combination | [64,85,149] | ↑ | ⟺ | ⇑ | - Magnitude of STL varies with different materials and combinations. |
| | Different Composite plate/shell types | [69,71] | ↑ | ⟺ | ⇑ | - Magnitude of STL varies with different composite materials. |
| | Different FGM plate/shell types | [150] | ↑ | ⟺ | ⇑ | - Magnitude of STL varies with different functionally graded materials (FGMs). |
| Orientated | Incident/elevation angles | [60,69,145] | ↓ | ⋯ | ↓ | - Poor STL performance. |
| | Azimuthal angles | [72,100] | ↓ | ⟺ | ↓ | - STL Increased slowly at mid-high frequency zones. |
| | Core inclination angle | [51] | ↑ | ↓ | ↓ | - Poor STL performance. |
| | Temperatures | [72] | ↓ | ↓ | | - Decreased STL. |
| | Stacking sequences | [49,68,69,71] | ⋯ | | | - STL improves with more zero plies. |
| | Cone angles | [145] | ↓ | ⟺ | ⋯ | - STL poorly increase at mid frequency region. |

↑: Increased STL at low frequency zone; ⟺: Increased STL at mid frequency zone; ⇑: Increased STL at high frequency zone; ↓: Decreased STL; ⋯: No significant difference in STL.

## 5.2. Optimization of Sound Transmission Loss of Double Panel Structures

Lightweight sandwich and composite double panels may suffer from lower STL especially in the mass-controlled region due to the effects of mass law. However, by correct optimization problem, the best solutions of reducing this challenge can be achieved. Moreover, design and shape optimization

has been a veritable method used by researchers to produce better performance of structures [151,152]. The optimization problem generally involves, for example,

finding the design vector $x$
that minimizes the objective function $f(x)$
subject to: constraint 1 (*inequality*), constraint 2 (*equality*)

To maximize the objective function, for instance, the STL, the minimized objective function can be negated. Optimization techniques can be applied, therefore, on these structures to maximize their STL and reduce their weight as demonstrated by Shojaeifard et al. [153] who used a genetic algorithm approach. Contemporary researchers have adopted the multi-objective optimization problem which involves the solution of a number of parameters capable of minimizing the objective functions and satisfying the sets of defined restrictions, or constraints. Solving a multi-objective optimization problem give rise to what is known as Pareto optimal solution. These are non-dominated vectors on the Pareto optimal set when the objective functions are plotted. In other words, at least one of the objective values must be degraded before the solutions can be improved in any of the objective functions. An optimization problem of panels with multiple layers and multi-objective criteria was undertaken by Tanneau et al. [154]. The combination of the genetic algorithm and the transfer matrix method was adopted by the authors to obtain the best solutions of the problem. In recent times, Talebitooti et al. [34] combined both the non-dominated sorting genetic algorithm and the first order deformation theory to obtain the optimal solutions of the STL of composite cylindrical shell with porous materials. Their optimization involved the use of first cost function to minimize the weight; and a second cost function was minimized to maximize the STL. Moreover, in their optimization procedure, two points were selected such that the first had a maximum STL and the second had a minimum weight of the Pareto front. The design variables were ply orientation, composite material types and porous material types. The optimized results due to ply orientation were compared with the non-optimized ones. Their results showed an increase in the STL for the two optimal points compared to the non-optimum one, in all frequency regions. A bi-optimization problem that involved maximizing the transmission loss and minimizing the weight was also carried out by Zhou et al. [155] and they obtained the Pareto fronts. In their result, the transmission loss increased at the expense of increased weight below the limit value of the Pareto front while above this front, the transmission loss decreased. One of the advantages of carrying out this bi-optimization problem by the authors is that it helped to distinguish the acoustic superiority of different structures. Optimization problems especially in the low frequency range have also been carried out by Zhang and Du [45] and Denli and Sun [156]. Core optimization of sandwich DPS in a wide frequency range has also been performed by [157]. Finally, Tsai et al. [158] adopted a conjugate gradient optimization technique to calculate and optimize the material properties on the STL of composite plate.

## 6. Summary and Conclusions

A critical overview of the sound transmission loss of sandwich and composite double panel configurations investigated by several contemporary researchers has been presented in this review article. The STL as a vibro-acoustic index has been used to measure the potential of materials to absorb sounds transmitted through them. Single and double panel structures made from different materials have been used as noise reduction device in many novel engineering applications such as in the transportation, building and construction industries. However, double wall structures have gained popularity over their single panel counterparts owing to their wider applications and better transmission loss behavior. The vibro-acoustic characteristics, especially the STL of sandwich and composite double panel structures have been studied by an appreciable number of researchers. The main goal is to reduce the amount of sounds transmitted in such materials by investigating various parameter effects on their STL. In this review, the influence of these parameters on the STL of sandwich and composite double panel structures under acoustic effect has been critically examined. The different

controlled regions in the STL versus frequency curve have been used to explain the effectiveness of the material to reduce transmitted sound waves.

This review has attempted to classify the parameter effects on the STL as geometrical, material, structural and orientated parameters. It is revealed that geometrical parameters such as increasing plate/shell thicknesses, core heights, core radii, plate sizes, finite and infinite dimensions of the panels etc., have significant influence on the STL. The material parameters such as increasing Poisson ratios, shear modulus, damping loss factors etc., however, have little or no effects on the STL. Orientated parameters such as incident, azimuthal, inclination angles, stacking sequence etc., have varying significant effects on the STL of the structure. The structural parameters which include core types, metal or composite plate/shell types etc., have been shown to have great influence on the STL. This indicates more prospects for future research in adopting different material combination for enhancing STL. This idea, for example, was demonstrated by Fu et al. [149] who attempted to combine the advantages of FGM carbon nanotubes by making them laminated composites sandwich plates. Their salient findings indicate an increase in the STL of this kind of material compared to the conventional ones. In spite of the promising prospects of enhancing the STL by using the FGM, the investigation of this type of material, however, is still in its nascent stage. Therefore, research efforts are needed to investigate more of the potentials of using this material type. In addition, the geometrical configuration of sandwich and composite DPS that have been mostly adopted for STL application are rectangular flat plates/shells and cylindrical or curved shells. More studies should be carried out to analyze and compare these topological effects on their STL using the same dimensional and parametric conditions. However, Arunkumar et al. [70] investigated the effects of three core topologies such as cellular core, trapezoidal core and triangular core, on their STL in the low frequency domain. In their results, it was observed that the STL was largely influenced with the different core topologies. Therefore, a quantifiable approach and shape optimization procedure can be performed to calculate the optimal STL of various geometrical shapes of both the plates and core.

Quite a good number of researchers have endeavored to use analytical and experimental methods to model and validate, respectively the STL of flat and cylindrical double panel shells/plates. The analytical procedures give the exact solutions while the measured experimental results give the actual solution of the physical problem. However, a major drawback of these two methods is that complex scenarios which may ensue in practical applications may be intractable. Numerical approaches, for example, finite element approximations are veritable means to addressing this drawback. For this purpose, more numerical procedures are needed not only to handle the complexities of the structure but also to virtually validate results obtained through analytical and experimental methods.

As has been previously mentioned and well reported that increase in the thicknesses and sizes of the composite and sandwich double plates or shells result in the increase in the STL. However, the increased parameters create additional weight and therefore elongate the computational period and final cost of production. While the utilization of lightweight sandwich and composite panels has promising potentials to overcome these drawbacks, however, their STL are mostly enhanced in the low frequency domain or before the mass controlled region. To adequately address these challenges, studies have shown that well defined optimization problems, for example, a multi-objective optimization problem, can be adopted. This solution and other optimization methods can maximize the STL in all frequency domains and equally reduce the weight of the material. However, very few research works have been carried out using different optimization problem procedures. It is therefore expected that more efforts should be channeled in this area because it promises a veritable solution to addressing many of the drawbacks encountered with enhancing the STL of these systems. In addition, with this approach, the design of sandwich and composite double panel structures can be tailored to specific industrial applications for optimal transmission loss results.

**Author Contributions:** Conceptualization, C.W.I., M.P. and S.W.; methodology, C.W.I., software, C.W.I.; validation, C.W.I., M.P. and S.W.; formal analysis, C.W.I., M.P. and S.W.; investigation, C.W.I., M.P. and S.W.; resources, C.W.I., M.P. and S.W.; data curation, C.W.I.; writing—original draft preparation, C.W.I.; writing—review and editing,

C.W.I., M.P. and S.W.; visualization, C.W.I.; supervision, M.P.; funding acquisition, M.P.; project administration, M.P. All authors have read and agreed to the published version of the manuscript.

**Funding:** The work presented here has been funded by the National Science Centre, Poland with decision no. DEC-2017/25/B/ST7/02236.

**Conflicts of Interest:** The authors declare no conflicts of interest.

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
