# Peer review of "Comparative Study of Sound Transmission Losses of Sandwich Composite Double Panel Walls"

_applsci, doi:10.3390/app10041543_

Round 1
Reviewer 1 Report
The paper deals with a comparative study of the STL in sandwich, flat, cylindrical or curved walls. The paper is quite interesting and well structured. Minor suggestions are proposed to improve paper’s quality in terms of coherence and readability.
Specify the numerical methods referred to in "2.1 Double panel air cavity structure". Can an analysis with different geometric conformations and / or constraint of the slab make sense? 15/47 - Negative Mach number? Please explain. The article is named as a comparison. However, this comparison is not so obvious but it is more of a list. Can it make sense to make a comparison between the forecast criteria, as well as the influence of the different parameters?
Author Response
Reviewer's comments
The paper deals with a comparative study of the STL in sandwich, flat, cylindrical or curved walls. The paper is quite interesting and well structured. Minor suggestions are proposed to improve paper’s quality in terms of coherence and readability.
Specify the numerical methods referred to in "2.1 Double panel air cavity structure". Can an analysis with different geometric conformations and / or constraint of the slab make sense? 15/47 - Negative Mach number? Please explain. The article is named as a comparison. However, this comparison is not so obvious but it is more of a list. Can it make sense to make a comparison between the forecast criteria, as well as the influence of the different parameters?
Authors' response
The authors would like to thank the reviewer for his comments. The numerical method is the finite element method which has been specified both in the description of Section 3.1, paragraph 1; and in Figure 2 of the revised manuscript. Yes, different configurations and boundaries can significantly affect the STL. The entire manuscript gives both a critique and logical comparison of different forms of double panel air cavity and core structures. For example, see Figure 3 and Section 5.1 of the revised manuscript. The influence of different parameters has been summarized in Table 2 of the revised manuscript.
Reviewer 2 Report
This paper presents an overview of the recent studies that focus on improving the acoustical performance, particularly sound transmission loss (STL), of sandwich and composite panels. The area of panel STL is well-established and has been extensively worked on over the past decades. This paper is timely in providing a recap on the topic and its recent advances. However, considering how well-established this topic is, the amount of discussion on the past works that lead to the recent works may be inadequate in the present form. In general, as a good review paper, works showing impactful findings should be discussed in detail, including complimenting the discussion with extracted figures. Additionally, for each new type of panel that is being discussed, a schematic diagram is usually provided for easier understanding of how typical designs are like (something like that shown in figure 6). It is unclear that whether the figures given in this paper are the work of the authors or are extracted from the literature. As such, at times, it was confusing trying to figure which is which. In summary, the purpose of the paper is clear but the conclusion section does not seem to align well with the purpose. After reading the paper, I do not understand the need for this review and how it could address the research gap. I recommend a major revision to consider the above and the following comments.
The structure of this paper generally lacks proper paragraphing. Consider improving it. In line 49, the acronym “STL” should be stated in line 44 instead where the term is first introduced. In line 64, shouldn’t the mass law be proportional to the given parameters rather than inversely proportional? In figures 1 and 3, it may be hard to visualise the two DPS configuration. Consider adding a schematic diagram of each DPS type within the respective graphs for easier understanding. In figure 2, the image does not provide any sense of size. Consider providing dimensions in the caption or in the main text. In figures 2 and 3, citation to the images should be provided. In figure 2, it is unclear where the incident and the transmitted sides are. Consider adding labels to indicate the sides. In figure 3b, the lines cannot be differentiated in greyscale printing. Consider adding line variation. In figure 3, the image appears to be truncated. Consider revising it. The discussion of validation is important but Section 3.2 as a standalone may not be able to communicate the purpose clearly. Most published works are generally validated in some way. Hence, it may be better to combine Section 3.2 with Section 3.1. Do the mathematical symbols correspond to the symbols shown in figure 6? What are o and p in equation 16? Is case study 2 referring to the same image shown in figure 6? In equations 33 and 42, shouldn’t the term inside log be an absolute value. If yes, the absolute sign should be indicated. In line 962, it is claimed that the pyramidal core gives the highest STL. This claim should be supported qualitatively. What is the justification that the pyramidal core is the best? In figure 8, are the results new data or are they extracted from the literature? If it is the latter, citations should be provided. Table 2 is quite well-done because it provides a clear overview of how the different parameters would affect the STL at low-, mid-, and high-frequency range. However, the symbols used to represent the level from 1 to 5 may be confusing. Consider changing the symbols to those that may be easier to understand. For example, the authors could consider using up and down arrows. Part of table 2 in page 36 is truncated. This truncation should be addressed.Author Response
The structure of this paper generally lacks proper paragraphing. Consider improving it. In line 49, the acronym “STL” should be stated in line 44 instead where the term is first introduced. In line 64, shouldn’t the mass law be proportional to the given parameters rather than inversely proportional?
The authors have adjusted the paragraphs in the revised manuscript as suggested. The acronym of the STL and definition has been stated accordingly as suggested. Thank you for the observation of the definition of mass law in line 64, the word ‘inversely’ has been removed.
In figures 1 and 3, it may be hard to visualise the two DPS configuration. Consider adding a schematic diagram of each DPS type within the respective graphs for easier understanding. In figure 2, the image does not provide any sense of size. Consider providing dimensions in the caption or in the main text. In figures 2 and 3, citation to the images should be provided. In figure 2, it is unclear where the incident and the transmitted sides are. Consider adding labels to indicate the sides. In figure 3b, the lines cannot be differentiated in greyscale printing. Consider adding line variation. In figure 3, the image appears to be truncated. Consider revising it.
Schematic diagrams have been included in Figures 1 and 3 as suggested. The material parameters and dimensions of the DPS used for the numerical simulation have been provided in the text (see Section 3.1, first paragraph). Figures 2 and 3 are the authors’ simulation results. Direction of the normal plane wave has been added to show the source of incident sound wave. The legend and truncated images have been addressed. Please, see the revised manuscript.
The discussion of validation is important but Section 3.2 as a standalone may not be able to communicate the purpose clearly. Most published works are generally validated in some way. Hence, it may be better to combine Section 3.2 with Section 3.1. Do the mathematical symbols correspond to the symbols shown in figure 6? What are o and p in equation 16? Is case study 2 referring to the same image shown in figure 6? In equations 33 and 42, shouldn’t the term inside log be an absolute value. If yes, the absolute sign should be indicated.
The authors wish to thank the reviewer for the suggestion of combining Section 3 of the previous manuscript. Both the methodologies and models for obtaining STL in DPS have been combined to give a continuous Section in the revised manuscript (Please, see Section 4 of the revised manuscript). However, the questions of mathematical symbols, equations 32 and 42 no longer apply. The Section related to these questions has been completely removed from the revised manuscript as suggested by another reviewer of the previous manuscript.
In line 962, it is claimed that the pyramidal core gives the highest STL. This claim should be supported qualitatively. What is the justification that the pyramidal core is the best?
The assertion of the statement that the pyramidal core showed the highest STL in all dip frequencies can be seen in Figure 8(a) (see below). From Figure 8(a), the pyramidal core showed the highest STL (followed by the 3D-Kagome core and then the tetrahedral core) in all peak and dip frequencies. Please note, due to inability to obtain copy right permission, the Figures under section 5.1 of the previous manuscript has been removed in the revised manuscript.
Figure 8(a) {previous manuscript}
Table 2 is quite well-done because it provides a clear overview of how the different parameters would affect the STL at low-, mid-, and high-frequency range. However, the symbols used to represent the level from 1 to 5 may be confusing. Consider changing the symbols to those that may be easier to understand. For example, the authors could consider using up and down arrows. Part of table 2 in page 36 is truncated. This truncation should be addressed.
The symbols have been modified as suggested. Please see Table 2 of the revised manuscript. Also, the truncation problems have been addressed in the revised manuscript.

Reviewer 3 Report
The manuscript is a review of the different double panel walls typologies and their effect on the sound transmission loss.
The topic is interesting from both an academic and industrial perspective. The concepts discussed in the manuscript are potentially worthy of publication.
However, the manuscript is not well organized and often lacking of clarity. The english language is often inadequate.
The focus of the discussion is also not very clear. The title suggests the manuscript is a review of the double panel walls technologies with the comparison of the respective performance, but then in Section 3 is presented a review of the characterization techniques and in Section 4 a review of the modelling techniques. The discussion then returns to the discussion of the double panel walls performance.
Section 3 and 4 seem out of place in the presented work. Nevertheless, those sections are interesting and contain useful information worth of publication, possibly in another work (after a revision of the English language, as stated before).
Here are more precise suggestions for the authors:
1) Section 1. Introduction should be revised:
Row 43-49: this is the definition of sound transmission loss, which is probably the most important definition of the manuscript, and is not clear. Please rephrase it. Add a figure if it helps.
Row 59-61: "... may have finite or infinite dimensions ..." This is not clear. What do you mean with "infinite dimensions"?
Row 75-134: all this part is useful for the discussion, but too specific for an Introduction. I suggest to shift this part in a new, specific section in which the STL curve is discussed in detail. It would be beneficial for the work to expand this part and made it much more clearer.
Table 1: please add the name of the frequencies in the table rows and add the definitions of the used parameters in the table caption.
2) Figure 1(a) "control" is missing in "coincidence control"
3) Section 2: please add a figure depicting the double panel structures ad the beginning of the section (or at the beginning of each specific subsection). This would greatly clarify concepts and improve the effectiveness of the discussion.
4) Section 3 and Section 4: I suggest not to discuss those topics in this work. Those sections are misleading and, even if very interesting, detrimental for the clarity of the presented work. Please consider to publish the content of those sections in another work and maintain the focus on the topic of this work.
5) Table 2: please consider using letters or arrows instead of the proposed arbitrary symbols in the table to improve its readibility. The used symbols carry no intrinsic meaning and are . Here a suggestion:
Increased STL: upright arrow ↑ Decreased STL: downright arrow ↓ No significant difference in STL: dash -
6) Overall, add references every time you assert something that is not obvious. For example, this occour in row 32, 40, 42, 43, 55, 172, etc.
7) Please improve the clarity of the discussion in all the sections.
Author Response
Section 1. Introduction should be revised:Section 1 has been revised as suggested. Please, see the revised manuscript
Row 43-49: this is the definition of sound transmission loss, which is probably the most important definition of the manuscript, and is not clear. Please rephrase it. Add a figure if it helps.
The definition of STL has been reordered and rephrased in the introduction of the revised manuscript.
Row 59-61: "... may have finite or infinite dimensions ..." This is not clear. What do you mean with "infinite dimensions"?
The sentence ‘…which may have finite or infinite dimensions’ has been removed. However, finite and infinite sizes of double panel structures and boundaries have been investigated by some researchers. (please, see References [] of the revised manuscript)
Row 75-134: all this part is useful for the discussion, but too specific for an Introduction. I suggest to shift this part in a new, specific section in which the STL curve is discussed in detail. It would be beneficial for the work to expand this part and made it much more clearer.
The suggestion to make the part a new section has been effected. Please, refer to Section 2 of the revised manuscript.
Table 1: please add the name of the frequencies in the table rows and add the definitions of the used parameters in the table caption.
The names of the type of frequencies and their corresponding definitions have been included in Table 1 of the revised manuscript.
Figure 1(a) "control" is missing in "coincidence control"The truncated word in Figure 1(a) has been addressed in the revised manuscript.
Section 2: please add a figure depicting the double panel structures ad the beginning of the section (or at the beginning of each specific subsection). This would greatly clarify concepts and improve the effectiveness of the discussion.Figures representing the double panel structures have been included. Please, see Figures 3, 4, 5, 6 and 8 of the revised manuscript.
Section 3 and Section 4: I suggest not to discuss those topics in this work. Those sections are misleading and, even if very interesting, detrimental for the clarity of the presented work. Please consider to publish the content of those sections in another work and maintain the focus on the topic of this work.Section 4 of the previous manuscript has been completely removed in the revised manuscript. However, Section 3 of the previous manuscript (now Section 4 of the revised manuscript) has been revised as suggested by another reviewer.
Table 2: please consider using letters or arrows instead of the proposed arbitrary symbols in the table to improve its readibility. The used symbols carry no intrinsic meaning and are . Here a suggestion:Increased STL: upright arrow ↑ Decreased STL: downright arrow ↓ No significant difference in STL: dash –
The suggestion of using arrow symbols has been effected. Please, see Table 2 of the revised manuscript.
Overall, add references every time you assert something that is not obvious. For example, this occour in row 32, 40, 42, 43, 55, 172, etc.Issues arising with references have been addressed in the revised manuscript.
Please improve the clarity of the discussion in all the sections.The entire revision process of the revised manuscript has ensured clarity of the paper. The manuscript has been proof read, edited and language fixed.

Round 2
Reviewer 2 Report
The authors have satisfactorily addressed all comments. I recommend the publication of the revised paper in Applied Sciences.
Reviewer 3 Report
The manuscript has been revised as suggested.
The English language style is acceptable, but can be improved. The phrasing makes the manuscript hard to read, and the excessive use of adverbs constantly breaks the discussion flow.
The manuscript is overall acceptable for publication.